# Predicting Gross Motor Function in Children and Adolescents with Cerebral Palsy Applying Artificial Intelligence Using Data on Assistive Devices

**DOI:** 10.3390/jcm12062228

**Published:** 2023-03-13

**Authors:** Lisa von Elling-Tammen, Christina Stark, Kim Ramona Wloka, Evelyn Alberg, Eckhard Schoenau, Ibrahim Duran

**Affiliations:** 1Center of Prevention and Rehabilitation, University Hospital, Medical Faculty, University of Cologne, 50931 Cologne, Germany; 2Department of Neurology, University Hospital, Medical Faculty, University of Cologne, 50931 Cologne, Germany; 3Department of Pediatrics, University Hospital, Medical Faculty, University of Cologne, 50931 Cologne, Germany

**Keywords:** cerebral palsy, gross motor function measure, GMFM-66, GMFCS, machine learning

## Abstract

Data obtained from routine clinical care find increasing use in a scientific context. Many routine databases, e.g., from health insurance providers, include records of medical devices and therapies, but not on motor function, such as the frequently used Gross Motor Function Measure-66 (GMFM-66) score for children and adolescents with cerebral palsy (CP). However, motor function is the most common outcome of therapeutic efforts. In order to increase the usability of available records, the aim of this study was to predict the GMFM-66 score from the medical devices used by a patient with CP. For this purpose, we developed the Medical Device Score Calculator (MDSC) based on the analysis of a population of 1581 children and adolescents with CP. Several machine learning algorithms were compared for predicting the GMFM-66 score. The random forest algorithm proved to be the most accurate with a concordance correlation coefficient (Lin) of 0.75 (0.71; 0.78) with a mean absolute error of 7.74 (7.15; 8.33) and a root mean square error of 10.1 (9.51; 10.8). Our findings suggest that the MDSC is appropriate for estimating the GMFM-66 score in sufficiently large patient groups for scientific purposes, such as comparison or efficacy of different therapies. The MDSC is not suitable for the individual assessment of a child or adolescent with CP.

## 1. Introduction

Cerebral palsy (CP) is an umbrella term for a heterogeneous group of motor disorders of the developing child [1]. The disorder of movement and posture is due to non-progressive damage of the developing brain and is the most common cause of motor impairment in childhood [1,2]. The symptoms of CP are highly variable [3]. The typical symptom in most cases is increased muscle tone, but can also be expressed as dystonia, chorea, or athetosis depending on the localization of the damage [4,5]. Neurological symptoms lead to abnormal posture and movement. Secondary skeletal deformities such as clubfoot, scoliosis, and hip dislocations may occur leading to pain [5]. In addition to motor impairment, there is a wide range of associated disorders, such as epileptic seizures [5] and sensory, behavioral, perceptual, cognitive, and communication disorders [1]. The clinical presentation and the severity vary from very mild limitations to severe physical disability. By distribution of the symptoms, CP is classified clinically into bilateral spastic CP (quadriplegic or diplegic), unilateral spastic CP, dyskinetic CP, ataxic CP, or mixed-type CP.

Due to the high complexity of CP, multidisciplinary care is required [6]. There is a broad field of therapeutic options, which are as diverse as the presentation of CP [7]. The evidence base of interventions continues to grow and there is a large increase in research findings [8,9]. Nevertheless, evidence for efficacy is often lacking, especially for standard care therapies [8]. Even for therapies that are already well evaluated, there is a lack of long-term outcomes [8]. In addition, more research is needed to find the best possible frequency and intensity of intervention [9].

The benefit of therapy varies inter-individually, and it is usually difficult to find appropriate selection criteria. On the other hand, there are many databases, such as those of health insurance companies and health care providers from which a lot of information could be generated. However, these databases usually only contain records about the type of CP, therapies, and prescribed medical devices, such as walking aids or wheelchairs. Motor assessments, such as the most frequently used Gross Motor Function Measure (GMFM-66), are often not available. The GMFM-66 score is the most commonly used assessment of gross motor skills and is a standard element for evaluating the success of therapy [10]. In order to increase the usability of available records, the aim of this study was to predict the GMFM-66 score from medical devices used by a patient with CP.

Obviously, a child or adolescent using a wheelchair is more restricted in their mobility than one using crutches. The medical device used usually matches the patient’s impairment like a “negative image”. The approach of using an assistive medical device for classification is not new. The Gross Motor Function Classification System (GMFCS), a five-level clinical classification system to describe the gross motor function of children with CP, also utilizes the use of aids as a basis for classification into groups [11]. Consequently, in this study, we aimed to determine the accuracy of measuring the level of gross motor impairment in children and adolescents with CP based on patient data including information about their medical devices. For this purpose, we developed the Medical Device Score Calculator (MDSC).

## 2. Materials and Methods

### 2.1. Study Participants

This study is retrospective, nonrecurring, and cross-sectional. The sample consists of data from 1639 patients who were treated within the routine rehabilitation program “Auf die Beine” from 2006 to 2021 at UniReha GmbH, Centre of Prevention and Rehabilitation of the University Hospital of the University of Cologne, Germany. The program “Auf die Beine” is aimed at children and adolescents with movement disorders such as CP, offering holistic, intensified, and interdisciplinary forms of therapy [12]. Trained healthcare professionals scored patients with the GMFM-66 and the GMFCS.

All patients or their legal guardians were informed, and consented to the collection of data for research purposes. Data were recorded in the German clinical trial registry and can be found at www.germanctr.de (DRKS0001131) (accessed on 16 December 2022). Data collection and use were approved by the Ethics Committee of the University of Cologne (16-269). Only data collected initially upon entering the program were used. Other data from follow-up examinations were not included in this study. Inclusion criteria for this study were the diagnosis of CP, age 2 to 25 years, and at least 60 items completed and scored on the GMFM-66, resulting in the inclusion of 1581 patients.

### 2.2. Gross Motor Function Measure-66 (GMFM-66)

The Gross Motor Function Measure is an evaluative test for the assessment of motor skills originally consisting of 88 test items and was reduced to 66 items in a revised version [10]. It was developed based on the milestones of normal motor development in children. A five-year-old normally developing child should be able to complete all test items with the highest possible score [13]. Scores range from 0 to 100, with 100 representing the highest motor ability [13].

Each test item is scored from 0–3 points:
0 = does not initiate1 = initiates2 = partially completes3 = completes completelyNT = Not tested

The GMFM-66 is one of the most investigated clinical test instruments for assessing the mobility of children with CP [14]. However, it is also a key element of research, e.g., to validate new therapeutic procedures or to evaluate other assessment instruments [13]. Reliability, validity, and responsiveness to change have been demonstrated to a high degree [10,13,14]. Scoring the GMFM-66 requires a computer program, the Gross Motor Ability Estimator (GMAE). It converts the entered scores into an interval scale.

### 2.3. Gross Motor Function Classification System (GMFCS)

The Gross Motor Function Classification System (GMFCS) is a standardized grading procedure for classifying the degree of physical limitation or motor performance of a child with CP aged 1 to 18 years [15,16]. The GMFCS consists of 5 levels for classifying the severity of CP, with level I representing the greatest level of ability and thus the lowest severity of the condition and level V representing the highest severity. An ordinal scale is used [11].

The GMFCS levels for children with CP (6 to 12 years) are [11]:Level I: Walking without limitations; limitations of higher motor skills.Level II: Independent walking without walking aids in the community.Level III: Walking with aids, limitations in walking in the community.Level IV: Independent locomotion limited; children are pushed in a wheelchair or use powered mobilityLevel V: Independent locomotion is severely limited even with powered mobility.

GMFCS grading is based on observation and/or report by the affected person or caregiver. Children are assigned to the level that best reflects their usual performance [11]. It does not assess the quality of execution or the best possible skill a child may be able to perform. The GMFCS primarily considers motor functions in the areas of locomotion, transfer, and sitting (trunk stability) [17]. The use of assistive devices represents a central element of the classification into a level. For each of the 5 levels, there are age-dependent defined descriptions of the characteristics.

### 2.4. Data Preparation

The previously collected data from the “Auf die Beine” program were anonymized and transferred in tabular form to the statistical software IBM SPSS Statistics. During this process, data were reviewed for correctness and missing data were added from existing medical records. In addition to the age, height, weight, and gender of the patients, the different assistive devices, the subtype of CP, and the GMFM-66 total score, subscores, and the GMFCS level were recorded. The different assistive devices were grouped appropriately based on their characteristics. After that, only aids or groups with a minimum count of *n* = 5 were used. Statistical analysis was initially performed using SPSS Statistics (Version 28.0.1.0).

### 2.5. Generation and Evaluation of the Four Machine Learning Algorithms

Machine learning is increasingly being used to process big data [18]. Used correctly, analyses of big data can quickly lead to research results [19]. In supervised learning, self-learning algorithms or neural networks are utilized to try to predict a known result as accurately as possible, based on various parameters [20]. Roughly, this procedure can be divided into two steps. In the first step (the feature selection), predictors or combinations of those are determined by their predictive power. In the second step, based on the feature expression, a function is searched that relates to the result [20]. For an optimal selection of an algorithm, it is advisable to use several algorithms in comparison [20].

For generating an algorithm for the Medical Device Score Calculator (MDSC), we compared four machine learning models in the software R (version 4.2.1.):Feed forward neural net (FNN)Random forest (RF)Support vector machine (SVM)Extreme Gradient Boosting (XGBoost)
with the software packages randomForest (version 4.7-1.1) [21], neuralnet (version 1.44.2) [22], nnet (version 7.3-18) [23], mlr (version 2.19.1) [24], and xgboost (version 1.6.0.1) [25].

In addition to the different assistive devices, the patient’s age and the type of CP (bilateral spastic, unilateral spastic, dyskinetic, ataxic, or mixed type CP) were included as variables. The different machine learning models all received the same input data with the aim to predict the GMFM-66 score. We omitted the GMFCS level as input data because it will generally not be available to health insurance companies. We were careful to use only data available to the health insurance company to predict the patient’s gross motor skills. The four models differ in the algorithm of how the prediction is calculated, with the XGBoost being a further development of the RF model. For a detailed description of the algorithms, please see specific literature [21,22,23,24,25].

The dataset was randomly selected and divided into two groups: The first group with 70% of the patients in the dataset was used to train the algorithms. The second group with 30% of patients was used afterward to predict GMFM-66 scores and the predicted Medical Device Score (MDS) was compared with the reported GMFM-66 scores. For this comparison, the concordance correlation coefficient (by Lin) (CCC), the mean absolute error (MAE), and the root mean square error (RMSE) were calculated. The concordance correlation coefficient (Lin) is a measure to assess the concordance of two measurement procedures performed on a patient (here GMFM-66 and MDS). The value ranges from 0–1, with a CCC of 0.61–0.80 indicating a strong concordance and a value > 0.80 indicating almost complete concordance [26]. The MDSC was also evaluated by using scatterplot and boxplot sorted by GMFCS level.

### 2.6. Evaluation of the Accuracy of MDSC in Group Analysis

Finally, analyses were performed to determine the accuracy of MDSC depending on group size in the analysis. To see if the MDSC can detect the true difference in GMFM-66 between two groups, different group sizes (30, 40, 50, 75, 100, 150, 200, and 250) were investigated. For this purpose, the study population (*n* = 1581) was randomly divided 10,000 times into 2 groups with 790 patients each. Of these, the divisions were chosen which differed most precisely by a certain point value (3, 4,…, 10). These two groups then formed the basis. From both groups, random samples were drawn in the above-mentioned group sizes. Then, it was considered whether a significant difference at level *p* = 0.05 could be detected between the samples of the two groups using the GMFM-66 and the MDS. This was repeated 10,000 times for each sample size. The ratio of samples that were significantly different and thus able to detect the true difference was expressed as a percentage.

## 3. Results

### 3.1. Descriptive Analysis

The study population included 1581 patients from the “Auf die Beine” program with an average age of 8.1 ± 4.3 years, with the youngest child being 2.2 years and the oldest patient 25.5 years old. Bilateral Spastic was the most common form of CP with a total of 75.6%. Most patients had GMFCS level III (37.4%) or IV (29.7%). A more detailed presentation is given in Table 1.

The type of medical devices used by the patients in the study is summarized in Figure 1. The figure shows the range of the GMFM-66 scores that were associated with the use of different medical aids. Additional bar graphs as an overview of the distribution of assistive devices at different GMFCS levels and types of CP are available in the Appendix A (Figure A1 and Figure A2, Table A1). The most frequently used assistive devices were transtibial orthoses (13.2%) and active wheelchair (12.8%); followed by posterior walker (9.5%), standing frame (7.6%), and night splints (6.6%).

In the descriptive analysis, the therapy bike with training wheels had the highest mean scores with a mean score of 71.19 GMFM-66 points followed by shoe inserts (61.06 points). This indicates that patients with higher gross motor skill scores are on average more likely to use those medical devices. The passive wheelchair (26.20 points) and the roller board (an aid for the initiation of crawling) (30.29 points) were reflected in the lowest GMFM-66 scores on average, which are therefore likely to be used more frequently by patients with low scores.

### 3.2. Evaluation of the Medical Device Score Calculator (MDSC)

Table 2 presents a summary of the evaluation procedure’s results referring to the prediction of the GMFM-66 score based on patient information and their medical aids by the four machine learning algorithms used in the comparison.

The results of all four algorithms showed strong accordance in the concordance correlation coefficient, with the support vector machine producing the lowest value of 0.72 (0.71; 0.78) and the others producing comparably average values [27]. However, the random forest method had the lowest mean absolute error and the lowest root mean square error and was thus used as the basis of the MDSC.

In the comparison of the GMFM score and the Medical Device Scores separated by GMFCS level in Figure 2, it is evident that MDS are overestimated for patients with higher GMFCS levels and are underestimated for patients with lower GMFCS levels. This scale shift can also be observed in the scatterplots (shown in Figure 3).

### 3.3. Results of the Accuracy of MDSC in Group Analysis

The results of the analysis of the power of the GMFM-66 and the MDSC, by means of the difference of the compared groups are shown in Table 3.

## 4. Discussion

The results of this study show the possibility to predict the motor function measured by the GMFM-66, based on the pattern of assistive device use, with an accuracy of a concordance correlation coefficient of 0.75 (0.71; 0.78) with a mean absolute error of 7.74 (7.15; 8.33), and a root mean square error of 10.1 (9.51; 10.8) applied to group comparisons. The Random Forrest model was found to be the most suitable of the four investigated machine-learning algorithms. However, the MAE of 7.74 (7.15; 8.33) of this model is not suitable to predict an individual GMFM-66 score because it is too high. In addition, it was shown that the MDS in patients with GMFCS levels II-IV correlated better with the GMFM-66 score than in patients with GMFCS level I or V. For patients with GMFCS level I, the GMFM-66 scores are more likely to be underestimated by the MDSC and for patients with GMFCS level V, rather overestimated (Figure 2 and Figure 3, “floor and ceiling effect”). The reason for this lack of correlation at the two ends of the scale could be the reduced need for medical devices for very severely or very mildly affected patients.

In order to evaluate the MDSC to detect a true GMFM-66 difference of, e.g., 5 points in two groups with a power of 80%, the sample size should be approximately *n* = 100 (in Table 3, *n* = 100 and 5 points gives a power of 79.5% for MDS). In contrast, for the GMFM-66 score used directly, the necessary sample size for detecting a 5-point difference in the GMFM-66 according to Table 3 would be approx. *n* = 75 (power of GMFM-66 for *n* = 75 and 5 points: 77.1%). This is only slightly higher than for the analysis using MDSC.

If a difference of, e.g., 8 points is to be recognized in the GMFM-66 scores in two study groups by MDSC with a power of 80%, then the sample size needed is *n* = 50. Therefore, the results of Table 3 can be used for sample size calculation in future studies that want to use the MDSC for predicting the GMFM-66 scores of a group.

The results above demonstrate the suitability of the MDSC to evaluate big data sets, such as those of health insurance companies. It is conceivable that using the MDSC, therapies can be retrospectively analyzed; not only longitudinally, but also across interventions comparing data sets including information about medical devices, but lack information about the severity of motor function skills such as a GMFM-66 score. To give an example, selective dorsal rhizotomy (SDR) could be considered: this is a surgical procedure on the posterior root of the lumbar spinal cord to reduce leg spasticity [28]. It would be possible to compare the MDS of children with SDR, calculated with the data of the health insurance companies, before surgery and, e.g., 2 years after surgery with a control group, which are of similar age and have a similar MDS at the beginning, in the sense of a case-control study. Accordingly, the MDSC could contribute to finding the most appropriate intensity and duration of therapy by a comparison of different therapies, and their durations and intensities. It could also be possible to assess the effectiveness of combinations of different therapies. Perhaps this type of analysis could identify patterns or groups that benefit more from specific therapies than others. By this means, therapies could be tailored more individually in the future and provide more benefits to the patient.

The disadvantages of estimating the motor function based on the supply of medical devices are that this observation only allows indirect conclusions about motor function. If, for example, the patients tend to have a supply of aids that is not in accordance with their needs, the motor function could be incorrectly predicted. Furthermore, medical devices that are procured outside of health insurance companies cannot be considered.

So far, only a theoretical/statistical evaluation has been done and it remains to be seen in application studies whether meaningful results can be generated using the MDSC.

It must be stressed that the MDSC does not seem to be suitable for an assessment of an individual. This measurement tool is not able to evaluate a single patient for clinical purposes with sufficient accuracy. It should only be used to assess groups of patients for research purposes, as described earlier. Expanding the size of the study population would likely generate even more accurate results. This is worth considering and striving for in the future to consequently obtain more accurate results.

### 4.1. Related Works

There are few studies that have used artificial intelligence to evaluate the gross motor function of children with CP. Duran et al. demonstrated that by using self-learning algorithms, the quantification of gross motor skills in children with CP could be made more efficient [12]. Zhang et al. used self-learning algorithms to assist in the assessment of gait analysis in children with CP [29]. To our knowledge, this is the first study that uses artificial intelligence to estimate gross motor skills based on patient data including medical device information.

### 4.2. Limitations of the Study

Study participant selection was determined by the participation in the rehabilitation program “Auf die Beine”. Children and adolescents with GMFCS Level I (7.5%) and V (7.1%) were particularly underrepresented because these two groups rarely participate in this rehabilitation program. In other studies, 34.2% and 15.2% of CP patients were classified as GMFCS Level I and V, respectively [30]. Thus, this selection bias resulted in bias. Furthermore, this study is based on a data set that was not specifically collected for this study and was examined retrospectively. The recording of the aids is not standardized. Therefore, completeness cannot be guaranteed.

## 5. Conclusions

The study results suggest that the MDSC is appropriate to predict differences in gross motor function in sufficiently large groups of children and adolescents with CP based on their medical device use for scientific purposes, such as comparison or efficacy of different therapies. The MDSC is not appropriate for the assessment of an individual child or adolescent in a clinical setting.

## Figures and Tables

**Figure 1 jcm-12-02228-f001:**
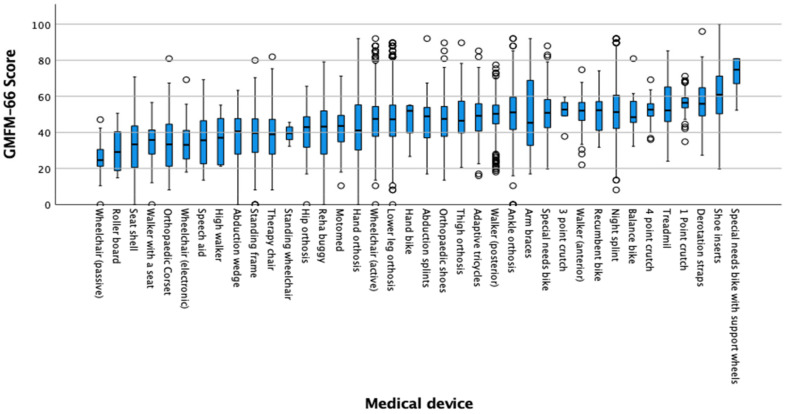
Statistical distribution of aids: Boxplot of the GMFM-66 scores of the different medical devices in descending frequency.

**Figure 2 jcm-12-02228-f002:**
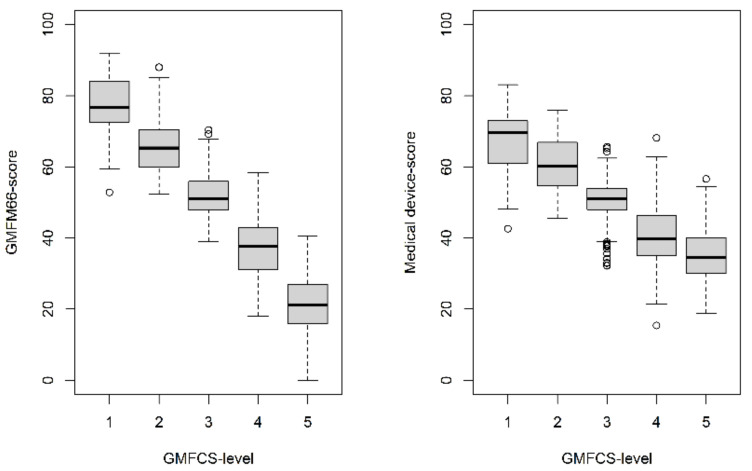
Comparison of GMFM-66 score and MDS by GMFCS-level.

**Figure 3 jcm-12-02228-f003:**
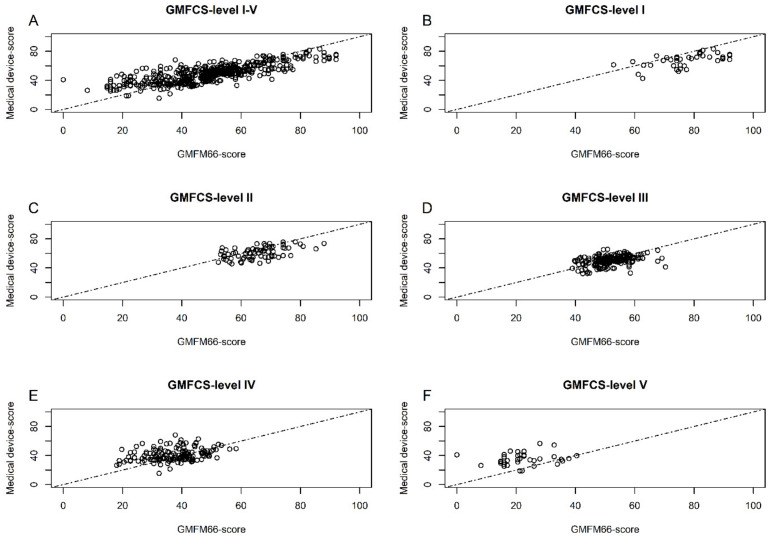
(**A**–**F**) Scatterplots with the unit line for the MDS against the GMFM-66 by GMFCS-level.

**Table 1 jcm-12-02228-t001:** Study population by GMFCS-level.

	GMFCS-level					
	I–V	I	II	III	IV	V
n	1581	118	289	592	470	112
Female, n	656	48	129	240	199	40
Age, years M (SD)	8.1 (4.3)	9.0 (4.3)	8.2 (4.2)	7.8 (4.2)	8.4 (4.5)	7.2 (3.6)
Height, cm M (SD)	119.1 (21.7)	130.2 (24.3)	122.3 (21.8)	117.5 (21.0)	118.4 (21.3)	110.6 (17.5)
BMI, kg/m^2^ M (SD)	16.0 (3.4)	16.7 (3.0)	16.3 (3.3)	16.3 (3.4)	15.7 (3.4)	14.3 (2.7)
CP subtype:						
Bilateral spastic, %	75.6	50.0	75.8	82.1	77.4	60.7
Unilateral spastic, %	7.0	39.8	12.1	3.5	1.5	0.0
Dyskinetic, %	5.8	2.5	1.7	3.7	7.7	22.3
Ataxic, %	2.0	2.5	4.2	1.7	1.1	0.9
Mixed type, %	9.7	5.1	6.2	9.0	12.3	16.1

Data are mean (M) with standard deviation (SD) unless otherwise indicated. Cerebral palsy (CP), Gross Motor Function Classification System (GMFCS), Body Mass Index (BMI). Bilateral spastic includes di- and quadriplegic types.

**Table 2 jcm-12-02228-t002:** Comparison of four different machine-learning algorithms to predict gross motor function using patient data and medical device information: Accuracy to predict the GMFM-66 using the MDS.

Algorithms	CCC	MAE	RMSE
RF	0.75 (0.71; 0.78)	7.74 (7.15; 8.33)	10.1 (9.51; 10.8)
SVM	0.72 (0.68; 0.76)	8.27 (7.63; 8.89)	10.8 (10.1; 11.5)
FNN	0.75 (0.71; 0.79)	7.86 (7.26; 8.46)	10.3 (9.67; 11.0)
XGBoost	0.75 (0.71; 0.78)	8.10 (7.50; 8.70)	10.5 (9.85; 11.2)

Data of the 95% confidence interval are given in parentheses; CCC concordance correlation coefficient (Lin), FNN feed forward neural net, MAE mean absolute error, RF random forest, RMSE root mean square error, SVM support vector machine, XGBoost eXtreme Gradient Boosting.

**Table 3 jcm-12-02228-t003:** Power of GMFM-66 and MDSC to detect differences in GMFM-66 score in samples of different group sizes with defined GMFM-66 score differences.

	Power of GMFM66-Score							
	GMFM66-Score Difference of the Compared Groups							
**Size of the Samples, *n***	**3**	**4**	**5**	**6**	**7**	**8**	**9**	**10**
30	0.0	0.5	2.0	3.9	20.2	68.1	**94.8**	**99.1**
40	0.1	1.8	5.5	20.5	71.3	**97.9**	98.8	100.0
50	0.6	4.4	12.8	58.8	**96.7**	99.9	100.0	100.0
75	3.6	16.1	77.1	**99.4**	99.9	100.0	100.0	100.0
100	10.9	60.9	**99.7**	100.0	100.0	100.0	100.0	100.0
150	53.3	**99.6**	100.0	100.0	100.0	100.0	100.0	100.0
200	**89.8**	100.0	100.0	100.0	100.0	100.0	100.0	100.0
250	99.8	100.0	100.0	100.0	100.0	100.0	100.0	100.0
	**Power of Medical Device-Score**							
	**GMFM66-Score difference of the compared groups**							
**Size of the samples, *n***	**3**	**4**	**5**	**6**	**7**	**8**	**9**	**10**
30	0.7	2.3	6.7	10.6	21.0	40.8	55.0	**80.6**
40	2.3	4.7	13.8	22.1	38.3	63.6	**82.0**	94.8
50	3.0	9.2	24.3	34.9	57.5	**80.4**	93.9	99.1
75	10.3	26.0	53.6	66.7	**88.2**	98.1	99.6	100.0
100	20.3	47.0	79.5	**88.4**	98.1	100.0	100.0	100.0
150	50.5	**82.1**	**97.9**	99.5	100.0	100.0	100.0	100.0
200	75.0	96.3	99.9	100.0	100.0	100.0	100.0	100.0
250	**91.0**	99.5	100.0	100.0	100.0	100.0	100.0	100.0

Bold is the smallest group size with which a difference could be shown in more than 80%. Of particular note, a maximum difference of 3 points was achieved in 89.8% using the GMFM-66 score with a group size of 200. Using the MDS, 250 patients are needed to achieve this for 91.0%.

## Data Availability

The dataset is available on request.

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
