# Peer review of "Predicting Gross Motor Function in Children and Adolescents with Cerebral Palsy Applying Artificial Intelligence Using Data on Assistive Devices"

_jcm, 2023, doi:10.3390/jcm12062228_

Round 1
Reviewer 1 Report
In this study, retrospective cross-sectional data from nearly 1600 children and young adults with cerebral palsy (CP) are analysed to determine with what accuracy it might be possible to predict gross motor function level (as measured by the GMFM-66) from the types of mobility devices used and a few other basis variables (age, type of CP).
In my comments below, I request clarification on a few points. I hope these comments will help the authors revise the paper for publication.
1. I’m not sure that the aim is precisely expressed.
- The paper says, “we aimed at developing effective tools that would allow accurate prediction of level of impairment and change of level of impairment using data about assistive devices used by CP patients”. I think that the aim is rather to determine the accuracy of using data about mobility devices to measure level of gross motor impairment in children, adolescents, and young adults with CP.
- The paper also says, “The long-term aim is to evaluate the effectiveness of therapies based on datasets that lack information about levels of impairment.” But the methodology of this study doesn’t allow the authors to address this aim, as the data are not linked to the therapies being delivered. This idea would be better reserved for the Discussion—where I see that it has been considered.
2. Throughout, the paper refers to the devices as “assistive devices” or “medical devices”. I think it would be more precise to call them “mobility devices,” as children with CP use other assistive and medical devices not included in this study (e.g., communication devices, hand orthoses, devices for sleep).
3. In Section 2.5, is it possible to describe the difference between the 4 models being compared: “feed forward neural net (FNN), random forest (RF), support vector machine (SVM), Extreme Gradient Boosting (XGBoost)”? Without any such explanation, it is not possible for a reader to interpret Table 2.
4. Was any sample size calculation performed to determine whether the study had sufficient power for the analysis?
5. I have difficulty understanding the statement that “two random groups of 790 patients each were generated from the data set n=1581 without replacement, so that both groups differed by, for example, 3 points in the GMFM-66 score”. As 790 is half of 1581, it sounds like the total sample were divided randomly into two groups. But then, if the allocation to the 2 groups was random, how did you ensure that they differed by 3 points on the GMFM? I think I must be misunderstanding something here. Could you please clarify?
I understand the next 2 sentences, but the following sentence doesn’t seem grammatical and I can’t make out the sense of it: “Then, it was considered whether the GMFM-66 score respectively the MDS of both samples differed significantly at level p = 0.05.” Could the authors please clarify? In the last sentence of this paragraph: “Samples found to be significantly different were expressed in percentage terms.” I’m not sure what the samples were significantly different from, or what the percentages mean. But perhaps that will become clear when the previous 2 sentences are clarified. As a result of my difficulties understanding this paragraph, I cannot interpret Table 3 in the Results section. More explanation of Table 3 might be needed.
6. In the Introduction, the authors note that “the Gross Motor Function Classification System (GMFCS), a five level clinical classification system to describe the gross motor function of children with CP, also utilizes the use of aids as a basis for classification into groups.” However, in the analysis (as I understand it), GMFCS was not used as a predictor of GMFM-66. It says, “In addition to the different assistive devices, the patients´age and the type of cerebral palsy were included as variables” but not GMFCS. What was the reason for that decision? Do the authors think that the inclusion of GMFCS might have improved the accuracy of the model?
7. The “roll board” is associated with low GMFM. Is this a device used for bed transfers? If so, do you think it might be helpful to the reader to say this in text, as it gets a special mention?
8. Table 1 appears to be a mixture of frequencies (n), means (with SDs), and percentages. Perhaps it would be clearer to give the total N at the top and then use only % and M(SD) in the body of the table? Please indicate clearly what is being presented in each row (whether n, % or M (SD)).
9. In Figure 1, “wheelchair (active)” appears twice. I think the one on the far left should say “wheelchair (passive)”.
10. In Section 3.2, the authors make the point that the two extremes of GMFCS levels are under- and over-estimated by the model. Is it advisable and possible to make adjustments in the model for individuals at GMFCS levels I and V to improve the accuracy of prediction?
11. In the first sentence of the Discussion, instead of “with an accuracy”, could you please state what level of accuracy? In more conventional analyses, I would want to work out a 95% CI, so that I could know how precise my estimates were. Can something like this be done for the analyses you present in this paper?
12. In the discussion, could the authors please comment on the possible risks of relying solely on data about devices to determine gross motor function? And also to determine response to therapies? They discuss the advantages of this approach, but I would like to see this balanced by a consideration of its disadvantages.
13. Under limitations, thank you for explaining the reasons for the unusual distribution across GMFCS levels in this sample. I was going to ask you about this. Reid et al (Using the Gross Motor Function Classification System to describe patterns of motor severity in cerebral palsy. Dev Med Child Neurol. 2011 Nov;53(11):1007-12) say that “For nine CP registries worldwide, the mean proportions of each GMFCS level, from level I to V, were 34.2%, 25.6%, 11.5%, 13.7%, and 15.6% respectively.” You may wish to cite this paper. (I’m not one of its author: I just think that it’s useful to have an international basis for comparison.)
14. In Table A1, please change the heading to say something like “GMFM-66 scores for individuals using the mobility devices”. These are not really “frequencies”, and the heading should indicate what the scores are (GMFM-66). Also in this table, please change the commas (,) to decimal points (.), as you have done for the tables in the body of the article. Also, do you need both SD and variance, or do you think SD is enough?
A FEW MINOR ISSUES:
Please change: “It is a non-progressive damage of the early developing brain” to “It is due to non-progressive damage of the early developing brain”. Or “caused by”. CP isn’t the damage itself but the result of the damage.
Please change: “and the insoles of 61.06 points” to “followed by shoe inserts (61.06 points)” to indicate that the shoe inserts had the second highest GMFM scores, and also to maintain consistency of terminology between Table A1 and the text).
Please change “great motor skill scores” to “gross motor skill scores”.
Please change “averagely” to “on average”.
Reviewer 2 Report
Artificial Intelligence to predict gross motor function in children with cerebral palsy
The work propose to use AI to classify the level of gross motor function (GMFM-66 score) in children with cerebral palsy.
The authors have provided a detailed review on the need for such an automated system to predict the GMFM score.
The authors have provided detailed description of their study setup in their section 2.
Further, in their results section, they have provided a detailed analysis on their experimental results.
Shortcomings,
While the paper has been well constructed and writen,
there are few minor concerns,
1. while the authors have mentioned that the data is in tabular form in section 2.4, it is necessary to explain more details on the data.
For example, what are the features in the dataset. Total number of records used for training and the feature size of each training record?
What is the split for the training and testing?
Does each record signify an individual candidate? if no, how were the data split, based on the records or based on the participants?
2. Since the authors are using AI, it is necessary to address some of the related works in this area. For example, a quick search on google scholar showed that
there are numerous works that has used AI/Deep learning to evaluate gross motor function of children for various neurological conditions. This analysis is
done through various methods, such as questionnaires, camera based methods, gait analysis and so on. The paper needs to include detailed related works
on analysis of gross motor functions using AI and Deep learning.
Examples of related works that uses AI/Deep learning to evaluate motor functions in children for various conditions:
1. Gattupalli, S., Babu, A.R., Brady, J.R., Makedon, F. and Athitsos, V., 2018, June. Towards deep learning based hand keypoints detection for rapid sequential movements from rgb images.
In Proceedings of the 11th PErvasive Technologies Related to Assistive Environments Conference (pp. 31-37).
2. Duran, I., Stark, C., Saglam, A., Semmelweis, A., Lioba Wunram, H., Spiess, K. and Schoenau, E., 2022. Artificial intelligence to improve efficiency of administration of gross motor function assessment in children with cerebral palsy.
Developmental Medicine & Child Neurology, 64(2), pp.228-234.
3. Vukićević, S., Đorđević, M., Glumbić, N., Bogdanović, Z. and Đurić Jovičić, M., 2019. A demonstration project for the utility of kinect-based educational games to benefit motor skills of children with ASD.
Perceptual and motor skills, 126(6), pp.1117-1144.
4. Kidziński, Ł., Yang, B., Hicks, J.L., Rajagopal, A., Delp, S.L. and Schwartz, M.H., 2020. Deep neural networks enable quantitative movement analysis using single-camera videos.
Nature communications, 11(1), p.4054.
5. Zhang, Y. and Ma, Y., 2019. Application of supervised machine learning algorithms in the classification of sagittal gait patterns of cerebral palsy children with spastic diplegia.
Computers in biology and medicine, 106, pp.33-39.
It is recommended that the authors create a subsection/section to discuss the related works that uses AI to evaluate different forms.
Reviewer 3 Report
Through the "Medical Device Score", which was created based on the analysis of a population of 1581 children with cerebral palsy, this retrospective, nonrecurring, cross-sectional study seeks to estimate the GMFM-66 score from the utilized medical devices of children with cerebral palsy. I have some comments/suggestions that the authors may find helpful to improve the clarity and overall presentation of their study. Please find them below.
1. Introduction (Ln 33-35):- “The appearance of cerebral palsy is highly variable. Typically, it is characterized by altered muscle tone, usually spasticity, muscle spasms, involuntary movements, dystonia, chorea, ballismus, and athetosis. Check the accuracy of this statement. Involuntary movements are not typical characteristics in all forms of CP but rather features of the dyskinetic form. And more, dystonia, chorea, ballismus, and athetosis, all are forms of involuntary movements.
2. The research gap that drove the authors to think about the current study is not clearly pinpointed in the study background.
3. It would be intriguing if authors could spare space in the introduction to shed light on the assistive devices that children with CP frequently utilize.
4. Ln 60-63. The Gross Motor Function Classification System (GMFCS), a five-level clinical classification system to describe the gross motor function of children with CP, also utilizes the use of aids as a basis for classification into groups. The authors may need to create further context and background information on the conceptual basis of the tool they developed in this study so that readers can understand why it could be more advantageous compared to the GMFCS.
5. It's unclear what the study's relevance is. Authors should note any gaps in the body of knowledge and underline the significance of filling such gaps. Additionally, a stronger justification would lead to a more explicit declaration of the potential implications for practice, which is currently insufficient.
6. Is it feasible to go into greater detail on the requirements for study participants' inclusion? Were there any factors considered to exclude certain prospective participants?
7. You analyzed data from patients with cerebral palsy between 2 to 25 years, But the title gives the impression that the study is for children (≤18 years old).
8. Ln 143-144. The different assistive devices were grouped appropriately based on their characteristics. Later, only aids with a minimum number of n=5 were used. What is intended by this is beyond my comprehension. Why a minimum number of 5 was considered for the grouping?
9. Ln 188. Bilateral Spastic was the most common form with a total of 75.6%. What bilateral spastic refers to? Spastic diplegia, spastic quadriplegia, or both? Clarify, please.
10. Authors should include in-depth discussion emphasizing the interpretation of their results, stressing their relevance to practice, and offering their viewpoint on how intended professionals make use of these data.
11. By completing the article, I am not convinced that the study is going to give me any new information to influence my decision-making concerning the prescription of assistive devices for children with cerebral palsy.
Round 2
Reviewer 1 Report
This is the second round of review for this paper, which reports analyses of retrospective data from nearly 1600 individuals with cerebral palsy (CP) to find a model for predicting gross motor function level (as measured by the GMFM-66) using information about the medical devices they have.
The authors have very thoroughly addressed the queries I raised in the previous review. I particularly appreciated having Section 2.6 so clearly re-written, thank you.
I have only 2 minor outstanding points.
Table 1 is still a mixture of different statistics, so could you please state on each row of the table whether it’s n, M(SD), or %?
Thank you for clarifying that the figures in parenthesis in Table 2 were 95% CIs. I still think that the first sentence of the Discussion should be changed, as the phrase “with an accuracy” is too vague.
Author Response
This is the second round of review for this paper, which reports analyses of retrospective data from nearly 1600 individuals with cerebral palsy (CP) to find a model for predicting gross motor function level (as measured by the GMFM-66) using information about the medical devices they have.
The authors have very thoroughly addressed the queries I raised in the previous review. I particularly appreciated having Section 2.6 so clearly re-written, thank you.
I have only 2 minor outstanding points.
Table 1 is still a mixture of different statistics, so could you please state on each row of the table whether it’s n, M(SD), or %?
Thank you for clarifying that the figures in parenthesis in Table 2 were 95% CIs. I still think that the first sentence of the Discussion should be changed, as the phrase “with an accuracy” is too vague.
Reply by the authors:
Thank you again for your comments. We have revised the table and the text section:
“The results of this study show that it is possible to predict the GMFM-66 score based on the pattern of assistive device use with an accuracy of a concordance correlation coefficient of 0.75 (0.71; 0.78) with a mean absolute error of 7.74 (7.15; 8.33) and a root mean square error of 10.1 (9.51; 10.8) to apply this to group comparisons.”
GMFCS-level |
||||||
I-V |
I |
II |
III |
IV |
V |
|
n |
1581 |
118 |
289 |
592 |
470 |
112 |
Female, n |
656 |
48 |
129 |
240 |
199 |
40 |
Age, years M(SD) |
8.1 (4.3) |
9.0 (4.3) |
8.2 (4.2) |
7.8 (4.2) |
8.4 (4.5) |
7.2 (3.6) |
Height, cm M(SD) |
119.1 (21.7) |
130.2 (24.3) |
122.3 (21.8) |
117.5 (21.0) |
118.4 (21.3) |
110.6 (17.5) |
BMI, kg/m2 M(SD) |
16.0 (3.4) |
16.7 (3.0) |
16.3 (3.3) |
16.3 (3.4) |
15.7 (3.4) |
14.3 (2.7) |
CP subtype: |
||||||
Bilateral spastic, % |
75.6 |
50.0 |
75.8 |
82.1 |
77.4 |
60.7 |
Unilateral spastic, % |
7.0 |
39.8 |
12.1 |
3.5 |
1.5 |
0.0 |
Dyskinetic, % |
5.8 |
2.5 |
1.7 |
3.7 |
7.7 |
22.3 |
Ataxic, % |
2.0 |
2.5 |
4.2 |
1.7 |
1.1 |
0.9 |
Mixed type, % |
9.7 |
5.1 |
6.2 |
9.0 |
12.3 |
16.1 |
Data are mean (M) with a Standard deviation (SD) unless otherwise indicated. Cerebral palsy (CP), Gross Motor Function Classification System (GMFCS), Body Mass Index (BMI). Bilateral spastic includes di- and quadriplegic types.
We have moved table 1 up one paragraph because otherwise, we would have three graphs or tables in a row and this could be cluttered.
Reviewer 3 Report
The necessary adjustments and recommendations have been properly addressed. The manuscript is potentially publishable in its current state.
Author Response
Thank you.